# The Role of Bone Morphogenetic Protein Receptor Type 2 (*BMPR2*) and the Prospects of Utilizing Induced Pluripotent Stem Cells (iPSCs) in Pulmonary Arterial Hypertension Disease Modeling

**DOI:** 10.3390/cells11233823

**Published:** 2022-11-29

**Authors:** Anichavezhi Devendran, Sumanta Kar, Rasheed Bailey, Maria Giovanna Trivieri

**Affiliations:** 1Cardiovascular Research Institute, Icahn School of Medicine at Mount Sinai, New York, NY 10029, USA; 2Black Family Stem Cell Institute, Icahn School of Medicine at Mount Sinai, New York, NY 10029, USA; 3Department of Medicine, Cardiology Unit, Icahn School of Medicine at Mount Sinai, New York, NY 10029, USA

**Keywords:** pulmonary arterial hypertension, pulmonary vascular resistance, vascular remodeling, induced pluripotent stem cell, disease modeling, pulmonary arterial endothelial cells, pulmonary arterial smooth muscle cells, bone morphogenetic protein receptor type 2

## Abstract

Pulmonary arterial hypertension (PAH) is a progressive disease characterized by increased pulmonary vascular resistance (PVR), causing right ventricular hypertrophy and ultimately death from right heart failure. Heterozygous mutations in the bone morphogenetic protein receptor type 2 (*BMPR2*) are linked to approximately 80% of hereditary, and 20% of idiopathic PAH cases, respectively. While patients carrying a *BMPR2* gene mutation are more prone to develop PAH than non-carriers, only 20% will develop the disease, whereas the majority will remain asymptomatic. PAH is characterized by extreme vascular remodeling that causes pulmonary arterial endothelial cell (PAEC) dysfunction, impaired apoptosis, and uncontrolled proliferation of the pulmonary arterial smooth muscle cells (PASMCs). To date, progress in understanding the pathophysiology of PAH has been hampered by limited access to human tissue samples and inadequacy of animal models to accurately mimic the pathogenesis of human disease. Along with the advent of induced pluripotent stem cell (iPSC) technology, there has been an increasing interest in using this tool to develop patient-specific cellular models that precisely replicate the pathogenesis of PAH. In this review, we summarize the currently available approaches in iPSC-based PAH disease modeling and explore how this technology could be harnessed for drug discovery and to widen our understanding of the pathophysiology of PAH.

## 1. Introduction

Pulmonary hypertension (PH) is a heterogeneous disease that is associated with high mortality. It is characterized by elevated pressure in the pulmonary artery, with a mean pulmonary arterial pressure (PAP) at rest of ≥20 mm Hg, and manifests clinically with dyspnea (shortness of breath), fatigue, and presyncope/syncope (lightheadedness/fainting). [1,2,3,4]

Pulmonary hypertension can be classified into five different groups [3] based on the most probable etiology, co-morbid conditions, and hemodynamic parameters (Figure 1), the Group I, or pulmonary arterial hypertension (PAH), that includes the “familial” or hereditary forms of PAH (HPAH); Group II or pulmonary venous hypertension due to left heart disease; Group III or pulmonary hypertension due to lung diseases and/or hypoxia; Group IV or pulmonary hypertension due to chronic thromboembolic disease; and Group V PH with unclear and/or multifactorial mechanisms [5].

The arterial form of PH or PAH is a condition of the vessel wall that causes progressive narrowing of pulmonary vasculature, with intima and media hyperplasia and formation of the plexiform lesions that over time lead to increased pressure, resulting in right ventricular dysfunction and death from right heart failure [6]. It is estimated that nearly 100 million people suffer from PAH worldwide [7], with the disease being more common among women with a female–male ratio of 2.3:1 [8]. The majority of the heritable forms of PAH are caused by mutations in the type II receptor of the *BMPR2* gene, which belongs to the transforming growth factor beta (TGF-β) receptor signaling family; of those carriers, 20% are at risk of developing PAH [9] over their lifetime, suggesting that a combination of inherited predisposition and environmental and epigenetic modifiers are required to elicit the phenotype. The most common disease-related *BMPR2* mutations are frameshift or nonsense mutations, with the missense mutations being very uncommon. Mutations in other members of the TGF-β signaling cascade have also been implicated in the pathogenesis of HPAH. Among them, the activin receptors like type 1 kinases (*ACVRL1* or *ALK1*), endoglin (*ENG*), and genes related to the intracellular downstream components of the BMP pathway such as SMAD 8 and SMAD 9 [10,11] have been described. The group I and specifically the familial/hereditary form of PAH (HPAH) is the prime focus of this review and is discussed in detail in the coming sections of this paper.

## 2. Vascular Remodeling in PAH

Remodeling of the pulmonary artery is considered the hallmark feature of PAH [12]. The arterial wall is a three-dimensional complex that is capable of adapting to variable blood pressure, blood flow, and shear stress. It is composed of three different layers, viz., the adventitia (outer), the media (middle), and the intima (inner). The outermost adventitial layer consists of fibroblasts, media smooth muscle cells (SMCs), as well as layers of elastic fibers, while the innermost intima consists of a single coating of endothelial cells (ECs). Within each layer, there are extracellular matrix constituents such as collagen, elastin, fibronectin, and proteoglycans that support the vascular architecture, tensile stability, and elasticity that enable an organized cellular migration and proliferation. The vascular arterial remodeling (VAR) seen in PAH typically encompasses all three layers of the vessel wall and, in the most severe forms of PAH, results in the thickening of all layers of the vasculature with cellular hypertrophy/hyperplasia and enhanced accumulation of the extracellular matrix [13].

One of the pathognomonic features of PAH is the appearance of smooth muscle cells in the small outermost pulmonary arteries, usually within the non-muscular respiratory acinus (Figure 2A). The cellular mechanisms behind this remodeling are neither fully elucidated nor understood. Preclinical models have shown that fibroblasts from the adventitia are activated following a hypoxic trigger, leading to their proliferation and subsequent migration to the medial and intimal layers. These fibroblasts further differentiate into smooth muscle-like cells that gradually propagate and generate matrix proteins that deposit along the arterial walls [14]. In carriers of the BMP mutation or other forms of inherited PAH, this insult is presumed to trigger hypertrophy of the arterial walls (Figure 2B), followed by the development of neointima (Figure 2C) and accumulation of myofibroblasts and extracellular matrix between the endothelium and the inner elastic lamina as well as neovascularization. In addition, upregulation of matrix metalloproteinases (MMP-1 and MMP-9) has been observed and appears to promote migration of the adventitial fibroblasts [15].

The most advanced form of VAR exhibits plexiform lesions (Figure 2D) that consist of gromeruloid-like structures with hyperproliferative ECs, SMCs, and matrix proteins that eventually result in vascular occlusion. Somatic frameshift mutations resulting in a premature stop codon of the beta receptors of transforming growth factor (TGFβR2) have been identified in about 30% of plexiform lesions in IPAH patients [15]. In addition, 90% of ECs within the plexiform lesions, unlike normal ECs, do not express TGFβR2. This aside, inflammation is presumed to play a substantial role in the process of VAR, as illustrated in PAH, especially in systemic inflammatory conditions such as systemic lupus erythematosus, where treatment with immunosuppressive therapy has been shown to improve clinical outcomes [16]. In fact, it is estimated that up to 30–40% of patients with PAH from various etiologies have evidence of elevated plasma levels of circulating autoantibodies, pro-inflammatory cytokines such as interleukin 1 (IL-1) and interleukin 6 (IL-6) and chemokines such as fractalkine and MCP-1 [12]. Additionally, inflammatory cells such as B/T lymphocytes, dendritic cells, mast cells, and macrophages have been found in the plexiform lesions of patients with severe PAH, and the pulmonary vascular endothelium of PAH patients expressed high levels of chemokines such as RANTES and fractalkine [17].

Aside from inflammation, the abnormal function of potassium channels, with the decreased expression of vasodilators like nitric oxide (NO) and prostacyclin, and overexpression of vasoconstrictors such as endothelin-1 (ET-1) are believed to contribute to the dynamic vasoconstriction and to further modulate vascular tone, resulting in vascular remodeling [15]. Indeed, the basis for current PAH therapy includes the oral prostacyclin receptor (i.e., IP) agonist selexipag [18] and the prostaglandin analogues such as iloprost, treprostenil, and epoprostenil [19] that target the prostacyclin pathway to promote vasodilation via the production of cyclic adenosine monophosphate (cAMP) [20]. Moreover, the soluble guanylate cyclase stimulator riociguat [21] and phosphodiesterase type V inhibitors such as sildenafil [22], that target the nitric oxide signaling pathway, are also used to elevate the levels of cyclic guanosine monophosphate (cGMP), leading to vasodilation. This in turn will maneuver the action of endothelin-1 receptor antagonists such as bosentan, which target the endothelium receptor to eventually bring about vasoconstriction [23]. Although these therapies in combination have been able to improve life expectancy, they have not contributed toward changing the course of the disease pathogenesis.

## 3. The Genetics and Gender Prevalence behind Pulmonary Arterial Hypertension

In 2000, a few heterozygous autosomal mutations in the bone morphogenetic protein receptor type 2 (*BMPR2*) gene were determined to be the genetic basis for heritable PAH [24,25]. Following this discovery, several other mutations (up to 400) have been detected in PAH patients (Table 1), 75% of which being identified in families with PAH [10,26,27]. Pathogenic *BMPR2* mutations have been found in nearly all exons within the gene, including the ligand binding domain, the serine-threonine kinase domain, and the cytoplasmic end terminus [28]. About 70% of those *BMPR2* mutations are either non-sense or frameshift mutations that might lead to haploinsufficiency by nonsense-mediated mRNA deterioration. The residual 30% are associated with diminished BMPR2 trafficking to the plasma membrane or decreased kinase activity [28].

Although the *BMPR2* mutation is identified in almost 82% of HPAH cases, the disease is clinically manifest in only 20% of subjects [2]. Furthermore, the penetrance of the mutation is higher in women than men, with 43% of female as opposed to 14% of male carriers developing PAH during their lifetime [29]. This variable penetrance suggests that other additional epigenetic/genetic secondary triggers or stimulants might play a role in decreasing *BMPR2* expression and/or signaling resulting in PAH [30].

**Table 1 cells-11-03823-t001:** Details of the gene mutations reported to be associated with pulmonary arterial hypertension.

Gene Symbol	Gene Name	References
*BMPR2*	Bone morphogenetic protein receptor type 2	[24,25]
*ENG*	Endoglin	[31]
*ALK1*	Activin receptor-like kinase 1	[32]
*BMPR1B* (*ALK6*)	Bone morphogenetic protein receptor type 1B	[33]
*SMAD4*	SMAD family member 4	[34]
*SMAD9*	SMAD family member 9	[34]
*GDF2* (*BMP9*)	Growth differentiation factor 2	[35,36]
*CAV1*	Caveolin-1	[37]
*KCNK3*	Potassium two pore domain channel subfamily K member 3	[38]
*TBX4*	T-Box 4	[39]
*EIF2AK4*	Eukaryotic initiation translation factor 2 alpha kinase 4	[40]
*ATP13A3*	ATPase 13A3	[36]
*AQP1*	Aquaporin 1	[36]
*SOX17*	SRY-Box 17	[36]

SMAD: Sma mothers against decapentaplegic; SRY: sex-determining region Y.

## 4. Bone Morphogenetic Proteins (BMPs) and Their Receptors

The bone morphogenetic proteins (BMPs) are cytokines of the TGFβ superfamily. They coordinate the development and differentiation of bone and cartilages [41] and further promote the development of various cell lineages such as mesenchymal and epithelial cells in addition to playing vital roles in vertebrate embryogenesis, specifically mesoderm induction, limb progression, collagen synthesis, hematopoiesis [42], and initiation of cardiovascular development in vertebrates [43]. The bone morphogenetic proteins bind to two different categories of transmembrane receptors, viz., the type I and type II receptors. These receptors [44,45] (Figure 3), comprise a small extracellular domain, a transmembrane domain, an intracellular domain with serine/threonine activity, and a cytoplasmic C-terminal tail section [46]. There are seven type I and three type II receptors; the type I receptors are also known as activin receptor-like kinase-1 or ALK (Figure 3). The type I receptors can bind to both BMPs (ALK1, ALK2, ALK3, and ALK6) and TGFβ (ALK5). Unlike TGFβ-proteins, BMPs can bind to type I receptors in the absence of type II receptors. Of the three type II receptors, the BMPR2 specifically binds to BMPs, whereas the other two serve as ligand for BMPs, activin, and myostatin [46]. Furthermore, the ligation of the BMP promotes the association of type I and II receptor to initiate phosphorylation and subsequent stimulation of type I kinase through the constitutively active type II receptor kinases.

### 4.1. The BMP Type II Receptor (BMPR2)

The bone morphogenetic receptor II differs from other type II receptors by an extended carboxyl (C)-terminal sequence positioned next to the intracellular domain, encoded by exons 12 and 13, respectively (Figure 4 and Figure 5) [8]. The BMPR2 receptor has a long and a short form, with and without the C-terminal tail, and can be found in most cell types of the lungs, heart, kidney, and brain. The role of the long C-terminal tail of BMPR2 remains currently unexplained. It has been hypothesized that the C-terminal domain interacts with the cytosolic proteins, altering signal transduction by means of different SMAD-independent pathways like Notch, MAPK, p38, and other extracellular signal-regulated kinases [47,48,49,50,51]. The short splice variant of the BMPR2 results from the alternate splicing of exon 12, which leads to a premature stop codon within exon 13, thereby shortening the cytoplasmic domain. This short form retains the ability to bind SMADs in vitro [47].

The truncated transcripts are usually diminished by non-sense mediated mRNA decay (NMD), and as a result, the disease may either be due to a dominant negative mechanism (by stable RNA) or to haploinsufficiency (by NMD transcript). However, the remote tissue-specific *cis*-regulatory factors that usually regulate transcription factors and signaling molecules may play a role as well. For instance, mouse Bmp2, Bmp4, Bmp5, and Gdf6 are part of the BMP super-family and are well-known to be regulated by long-range regulatory sequences [52]. However, the role of BMPR2 alternative splicing or long-range regulatory sequences in determining penetrance of heritable pulmonary arterial hypertension remains unclear [53]. What is certain is that the downregulation of BMPR2 is critical, since impaired BMP-related signaling pathway is a characteristic mechanism of PAH, regardless of genetic pathogenesis and in the absence of the known disease mutations.

### 4.2. The Bone Morphogenetic Protein Receptor Signaling

The BMP2/4 interacts with ALK3 and ALK6 to establish an active receptor complex [54,55]. On the other hand, the BMP6/7 binds to ActR-IIA along with ALK2 and ALK3 [54,56,57], while BMP9/10 binds to BMPR2 together with ALK1/2 [58,59] and signals through ActR-IIA or endoglin, a co-receptor that encodes for the TGF-β/BMP signaling pathway [60] (Figure 3). The type II receptors, after binding with the BMP ligand, phosphorylates and thereupon activates the type I receptors, which sequentially phosphorylates the receptorreceptor-regulated SMADs such as SMADs 1, 5, and 8. The BMP-specific R-SMADs form a complex along with SMAD4, which translocate to the nucleus, where it associates with the transcriptional co-activators like p300, CBP, Runx2, and GCN5 or co-repressors like c-Ski, SnoN, Tob, and SIP1 [44,61,62], resulting in the activation or inhibition of gene transcription. Conversely, the TGF-β ligands signals through SMAD2 and 3, but also through ALK1 and 2, which further promotes the activation of SMAD 1 and 5 in endothelial and other cells [63,64].

The downstream targets of the BMPR2 pathway include signaling mechanism that aim to control pulmonary artery smooth muscle cell (PASMC) generation by regulating the cell cycle [51] as well as a number of transcription factors such as Hey1 and Tcf7 that are involved in Notch and *Wnt* signal transduction [46].

The BMP signal transduction pathway is tightly regulated at different stages, both temporally and spatially, by way of diverse intracellular and extracellular signaling [44,65]. This aside, the BMP extracellular antagonists include gremlin, noggin, chordin, follistatin, and others in the DAN family that can be up-regulated by BMPs, thereby resulting in negative feedback loops [66,67]. On the contrary, BMP1, kielin, or chordin-like protein (KCP) and BMP-binding endothelial regulator (BMPER) can promote the BMP signal transduction pathway [68,69,70].

At the level of the cellular membrane, BMP signal transduction is blocked by BAMBI (BMP and activin membrane-bound inhibitor), which inhibits the production of receptor signal complexes. In contrast, the membrane proteins of the repulsive guidance molecule (RGM) family that is attached to glycosylphosphatidylinositol (GPI) augments BMP signaling by establishing complexes with the BMP type I receptors [71]. Intracellularly, SMADs 6 and 7 function to antagonize the SMADs that bind to type I BMP receptors and thereby inhibits the activation of downstream SMAD1/5, which in turn negatively regulates BMP signal transduction [72,73].

In addition, BMP signal transduction is regulated by the ubiquitin-facilitated proteasomal degeneration of SMAD 1 and 5 via the E3 ubiquitin ligase Smurf, which interacts with the SMAD 6 to modulate the degradation of BMP type I receptors [74,75]. The other mode of regulation of BMP signal transduction pathway includes diverse microRNAs (miRNAs) like miR-21 and miR-302 [76,77], along with 12-kDa protein (FKBP12), which binds and interferes with the BMP type I receptors [78].

## 5. Challenges Involved in Investigating Pulmonary Arterial Hypertension

The limited availability of human tissues that can only be obtained from patients with end-stage PAH has hindered the current understanding of PAH disease development and progression. Moreover, animal models do not accurately reflect human disease conditions due to specific discrepancies (species) and distinct pathogenic mechanisms that generally fail to recapitulate the complex pathogenesis of subjects with pulmonary hypertension. For example, the characteristic dysfunction of PAECs in PAH has been demonstrated in cultured cells and animal models of PH. Nevertheless, the type and severity of PAEC dysfunction, aberrations in gene expression, and the response to specific treatment regimen may differ from person to person (personalized medicine). In addition, there is limited availability of patient-specific PAEC to investigate the onset of PAH. Most importantly, the present existing animal models of PH only reproduce some but not all features of the human pathology. As such, new alternative models are required to gain a better understanding of the underlying molecular mechanisms of disease initiation and progression and for the development of new targeted treatments. Induced pluripotent stem cells (iPSCs) have been utilized to model several human cardiovascular diseases, including PAH [79,80,81,82]. Recently, we have witnessed an increasing interest in using this tool due to the ability of the iPSCs to differentiate into all somatic cell lineages, facilitated by genome editing tools and model systems. The evidence supporting the usage of stem cells to model PAH is presented below.

## 6. Human Pluripotent Stem Cell (hPSC) Utility for Disease Modeling Systems

Human pluripotent stem cells (hPSCs) provide a novel model system to investigate the genetic origin of human cardiovascular diseases. Compared to other preclinical animal models, hPSCs can be tailored to patients of interest and share an identical genetic background. Additionally, because hPSCs can be differentiated into diverse cell types (i.e., cardiomyocytes and endothelial and smooth muscle cells), they constitute an infinite source of material for studying and unraveling the molecular basis of various human diseases (Figure 6).

The hPSCs exists in diverse forms. Human embryonic stem cells (hESCs) were first described in 1998 and were acquired from early-stage human embryos [83,84]. These cells can be efficiently differentiated into all cell types, including SMCs and ECs, but their utility is challenged by ethical uncertainties and as such are restricted, with limited accessibility.

The second form of hPSCs are stem cells that are obtained via somatic cell nuclear transfer (SCNT), where the nucleus from a differentiated cell is transferred into an enucleated ovum [85]. Using this strategy, a cloned sheep was first created in 1996. Additionally, hPSC lines were similarly generated from a patient with type 1 diabetes [86]. The advantage of these cells is that they are a genetic match to the donor, but similar to hESCS, their use for research purposes is restricted.

The third form of stem cells, the induced pluripotent stem cells (iPSCs), are the most common and most extensively utilized type of hPSCs. They were first described in 2006, when Yamanaka along with his colleagues demonstrated the reprogramming of non-germinal cells (somatosomes or autosomes) into pluripotent stem cells through the heterologous expression of a combination of transcription factors [87], well-known today as the reprogramming factors or the Yamanaka’s factors: Oct3/4, Sox2, Klf4, and c-Myc, [87]. Initially studied in mouse fibroblasts, this hallmark discovery was followed by a series of studies that validated the reprogramming ability of the Yamanaka’s factors in numerous human somatic cells [88,89,90,91]. The iPSCs can be differentiated into the three embryonic germ layers, viz., ectoderm, mesoderm, and endoderm [92](Figure 7), and can form teratomas after subcutaneous or intramuscular injection into immunocompromised mice [88,89,90,93]. At the cellular and molecular levels, iPSCs are characterized by the expression of (1) critical pluripotent markers or genes like OCT4, NANOG, and SOX2 [94], (2) embryonic markers like SSEA3, TRA-1-60, and TRA-1-81 [95], as well as (3) concurrent repression of lineage-specific genes characteristic of the adult cells from which the iPSCs were originated [96]; the iPSCs also exhibit regulatory telomerase expression [97]. When in culture, the iPSCs form cobblestone-shaped colonies, with distinct nucleoli and precise, discrete cell margins or edges [96].

Following this remarkable discovery, iPSCs have been effectively differentiated into numerous cell types such as neurons [98,99,100], blood cells [101,102], adipocytes [103], fibroblasts [104,105], endothelial cells [101,106], smooth muscle cells [107], and cardiomyocytes [108,109]. They have also been originated from subjects with a variety of monogenic diseases [90] such as HPAH [99], for which iPSCs have been used to generate pulmonary vascular cells, thus providing a new avenue for disease modeling.

In addition to recapitulating diseases in vitro, iPSCs are also pliable to genetic manipulation [110] with zinc-finger nucleases (ZFN) [111]), transcription activator-like effector nucleases (TALENs) [112], and CRISPRCas9 (clustered regularly interspaced short palindromic repeats CRISPR-associated protein 9) [113].

The CRISPRCas9 system is presently the most widely used strategy in gene editing and is both relatively inexpensive and easy to use. It introduces double-stranded DNA nicks at specific locations using targeted guide RNA. Using this tool, disease-related mutations [114] have been introduced but also corrected in patient-derived iPSCs [81], enabling the generation of isogenic lines that are genetically indistinguishable from the original line except for the newly inserted mutation. This is particularly relevant as it limits possible confounding factors from inter-line inconsistency and is especially important in the study of HPAH, where there is incomplete phenotypic penetrance in unaffected mutation carriers [104] and access to “*de novo*” mutants. As a result, the corrected isogenic lines become critical for investigating the exact and detailed function of the dysregulated BMPR2 signaling pathway. In other words, it could be a more precise model system for studying PH.

### 6.1. Induced Pluripotent Stem Cell Model Systems for Pulmonary Arterial Hypertension

The limited access to human tissues and the requirement of repeated biopsies to investigate individual *BMPR2* mutations has resulted in significant efforts to develop iPSC-derived stem cells that could serve as potential surrogates for PASMCs and PAECs.

West et al. [79] attempted to differentiate two wild-type and two PAH-patient-derived iPSC lines into vascular mesenchymal stromal cells (iPSC-MSCs) and endothelial-like cells (iPSC-ECs) and compared their gene expression profile by microarray to that of primary pulmonary arterial endothelial cells obtained from controls and idiopathic PAH patients. They investigated the effect of the BMPR2 signal transduction at different stages of differentiation in BMPR2 mutant iPSC-derived SMCs and iPSC-derived ECs and identified 33 and 18 dysregulated genes linked to developmental processes and apoptosis, respectively. Among these, the Wnt receptors such as FZD4 and FZD5 and Wnt-secreted modulators such as SFRP1 and SFRP2 were found to be upregulated in BMPR2 mutants compared to the control groups. They concluded that altered BMPR2 signaling resulted in augmented WNT/β-catenin signaling in the PAH line [79]. Supporting these conclusions but also validating the ability of iPSCs in recapitulating this aspect of the disease in vitro, increase in the expression of Wnt-related genes such as WuFZD4, FZD10, and AZIN2 was observed in lung tissues isolated from 22 IPAH cases (i.e., without any BMPR2 mutations) [115,116]. Nonetheless, compared to the primary cells, the iPSC-derived ECs and iPSC-MSCs did not have the same proliferative features and apoptotic activity [79]. Hence, additional investigations will be crucial in elucidating the link between decreased BMPR2, Wnt signaling, and apoptosis in relationship to PAH.

In addition to the iPSC-derived model systems for PAH described above, iPSC-derived ECs have been utilized as substitutes for PAECs to model PAH in two subsequent studies [80,81]. Sa et al. [80] obtained iPSCs from skin fibroblasts of HPAH and IAPH patients, differentiated them into iPSC-derived ECs, and compared them with PAECs acquired from the same patient to validate their ability to recapitulate different PAH-related disease phenotypes [80]. Compared to PAECs derived from IPAH and HPAH cases, those iPSC-derived ECs did not show enhanced proliferation, mitochondrial hyperpolarization, or DNA damage [80,117]. Conversely, characterization of iPSC-ECs and primary pulmonary artery endothelial cells (PAECs) revealed an identical predisposition to apoptosis and diminished BMPR2 signaling, cell adhesion, and tube formation in comparison to the iPSC-ECs and PAECs obtained from the controls [80]. Furthermore, the RNA-Seq analysis of these cells showed that reduced migration and survival resulted from elevated expression of kisspeptin 1 (KISS1) and the subsequent downregulation of carboxylesterase 1 (CES1) [80]. Moreover, IPAH- or HPAH-derived pulmonary arterial ECs and iPSC-ECs responded comparably to the immunosuppressant FK506 and the neutrophil elastase inhibitor, elafin, which equally promoted angiogenesis in the two patient-derived iPSC-EC and PAEC lines, respectively. 

This study further indicated that the loss of response to FK506 and elafin in the other cell lines that were tested could be attributed to the elevated expression of the anti-migratory factor split guidance ligand 3 (SLIT3), thereby implying that SLIT3 may be a crucial genetic regulator in PAH [80]. The same investigators further explored the role of additional genetic modifiers among three families with HPAH cases, and unaffected mutation carriers (UMCs) [71]. The UMC-derived iPSC-ECs showed increased cell adhesion and survival in comparison to HPAH-derived iPSC-ECs with the same *BMPR2* mutation. Specifically, there was increased expression of the BMPR2 activators and diminished expression of BMPR2 inhibitors among UMC-derived iPSC-ECs vs. HPAH-derived iPSC-ECs [71]. Cell survival was preserved in UMC-derived iPSC-ECs and appeared independent from BMPR2 signaling [71]. Conversely, UMC-derived iPSC-ECs compared to HPAH-derived iPSC-ECs had upregulation in the baculoviral IAP repeat covering 3 (BIRC3), a gene that is thought to defend against ECs apoptosis. Following CRISPRCas9-mediated correction of the C118W BMPR2 mutation in HPAH-derived iPSC-ECs, cell adhesion, cell survival, BMPR2 regulatory gene targets, and BIRC3 expression continued to be dysregulated, implying that BIRC3 expression was not dependent upon BMPR2 signal transduction [71]. Additional studies using loss of function experiments with siRNA-mediated BIRC3 knock-down resulted in enhanced apoptosis in UMC-derived iPSC-ECs, demonstrating how BIRC3 is important for cell survival and as a protective genetic modifier in PAH [71]. These results are consistent with earlier reports demonstrating how decreased BMPR2 signaling in PAECs is related to elevated apoptosis in response to injury [108] as well as reduced adhesion and migration [109].

In a recent study, Kiskin et al. [82] utilized iPSCs as surrogate of pulmonary artery smooth muscle cells and demonstrated how the *BMPR2* mutation was adequate to determine the PAH-associated cellular phenotypes observed in primary PSMC lines. The differentiation methodology used for the generation of iPSC-SMCs is expected to have a significant impact on modeling PAH using *in vitro* systems that combined ECs and SMCs within organoid structures or pulmonary artery-on-a-chip models [79].

Furthermore, while another group has been successful in generating iPSCs (*BMPR2*) from PAH patients, they are yet to elucidate in detail PAH molecular or functional pathogenesis using the iPSC model systems. [118]

### 6.2. Limitations for Currently Available iPSC Model Systems for Pulmonary Arterial Hypertension

Despite the success of those initial studies in “creating” an iPSC-based model that could recapitulate the aspects of PAH related disease phenotypes in vitro, including dysregulated signaling pathways, there are nonetheless several limitations.

#### 6.2.1. Gaps in Knowledge

The *BMPR2* is expressed throughout the developing fetus and is critical for proper function and development of the vascular system. It is widely present in the endothelium of the pulmonary arteries of the lung [111] and is known to play a critical role in mediating adaptive response of the pulmonary vasculature to various stimulating factors, both endogenous and exogenous. Nonetheless, being a carrier does not confer an absolute risk for the development of PAH. As such, prospective studies looking at the interplay between the molecular mechanisms of BMPR2 and other critical genetic mutations and epigenetic or environmental factors are essential.

Likewise, the impact of gender in *BMPR2* mutant carriers with/without PAH also needs to be further explored to understand why PAH is more prevalent among women and linked to decreased survival in male *BMPR2* mutant carriers [112,113]. Additional studies using iPSC-ECs and SMCs in groups other than HPAH are also warranted. This is particularly relevant given that treatment is available only for patients classified as Group I or IV PH [114] and we have no specific therapies for the other types of PH, especially group 2 PH, which is the most globally prevalent group. [92,115]

Lastly, although Sa et al. [70], Gu et al. [71], and West et al. [69] showed how iPSC-derived ECs could serve as alternatives or substitutes for PAECs, there remains very little understanding of the utility of iPSC-derived SMCs, their similarity to PAH-related pulmonary artery smooth cells, and their contribution to the disease phenotype. Hence, establishing standardized protocols for iPSC differentiation into vascular smooth muscle-like cells will offer additional avenues for a better understanding of the molecular cues behind the onset and progression of PAH.

#### 6.2.2. Gaps in Methodology

In vitro studies aimed at the identification of therapeutic strategies in PAH have been developed using conventional two-dimensional (2D) systems, but they do not recapitulate in vivo conditions for what pertains to cell–cell interactions, signaling, and tissue organization. As such, for iPSCs to become a model system for PAH or other vascular diseases, generation of humanized three-dimensional (3D) tissue-engineered pulmonary systems using iPSC-SMCs/ECs will be crucial. Additionally, we will need to create systems that address the interplay and crosstalk between cells, using co-culture of iPSC-ECs and other vascular cells [119] or cell types such as inflammatory cells and their secretory cytokines and chemokines. The iPSC-derived organoids (organ-like tissue) might have the potential to offer such a system and have been successfully used to create vascularized liver and kidney organoids [120,121]. Similarly, the development of lung organoids or pulmonary artery-on-a-chip strategies could deliver an innovative strategy to investigate the genotype–phenotype correlation associated with unraveling disease progression in PAH, but also provide the mechanistic insights into the critical role of mechano-biochemical signaling cues in PAEC–PASMC crosstalk on vascular remodeling. This in fact could be a limitation of this review paper, given the current difficulties in drawing conclusions on genotype–phenotype correlations. Indeed, novel micro-engineering strategies offer the great potential to “create” models that could provide and cater to either a single or multiple hypothesis for a given experiment or pathway. In the last few years, the concept of tissue-on-chip has stirred quite a lot of attention in the area of medical and experimental research pertaining to PAH [122]. Tissue-on-chips or tissue chips are microfluidic apparatuses with micron-sized conduits or networks that enable cells to be grown in a more biologically regulated setting. In a fairly recent study, Al-Hilal et al. [122] pioneered and conceived a tissue-on-chip model that mimicked a PAH-like vessel phenotype to elucidate the gender discrepancy in PAH and to further probe into their efficacy for drug discovery. Nonetheless, despite their immense potential, the PAH-on-a-chip strategy will require additional standardization to accomplish a near-perfect pulmonary vascular model system [123].

In summary, iPSC-derived models display significant potential for improving our understanding of the process of vascular remodeling seen in PAH and may eventually lead to the development of disease specific molecular therapies that could have dramatic impact in the field [124].

## 7. Emerging Therapeutic Modalities for Pulmonary Arterial Hypertension Targeting the *BMPR2* Signaling Pathway

The existing pharmacological therapeutic regimens for PAH (as discussed earlier in this review) are centered on four pathways: (i) endothelin (ET) (ET receptor antagonists), (ii) nitric-oxide–cyclic-guanosine monophosphate (NO–cGMP promoters), (iii) prostacyclin (PGI2) (PGI2 pathway agonists), (iv) calcium channel signaling pathways (calcium-channel blockers or simply CCBs) [125,126,127,128,129,130].

Presently, there are fourteen drugs that have been approved by the Food and Drug Administration (FDA) in the United States for the treatment of PAH and chronic thromboembolic pulmonary hypertension (CTEPH). Nonetheless, the mortality rate remains high, and there are no treatments that have shown to halt disease progression. As a result, there is an unfulfilled need for newer treatments that go beyond pulmonary vasodilation and target therapeutic remodeling. The potential novel therapeutic regimen that targets via the *BMPR2* are discussed below.

a.Targeted Delivery of Exogenous BMPR2

The replacement of defective BMPR2 with exogenous wild-type receptor to PAH-affected vascular cells was explored via nebulized intratracheal delivery of adenovirus expressing BMPR2 in monocrotaline-treated (MCT) rat model systems. This approach was able to target the pulmonary arteries but failed to alleviate PH [131,132]. Reynolds et al. further modified this delivery system to specifically target the pulmonary endothelium, utilizing a bispecific-conjugated antibody for the highly expressed angiotensin-converting enzyme (ACE). In both MCT-treated and chronic hypoxic rat PH models, they saw an increase in BMPR2 expression and phospho-Smad1/5/PI3K signaling and a reduction in the levels of TGFb and phospho-Smad3/MAPK, with decreased right ventricular systolic pressure (RVSP), RV hypertrophy (RVH), and vascular remodeling that led to improved cardiac output [132,133,134]. The same group conducted a pre-clinical cell-based study utilizing bone-marrow-derived endothelial-like progenitor cells (BM-ELPCs) isolated from rat femur bones that were transduced with the adenoviral BMPR2 [135]. These intravenously delivered BM-ELPCs were found in the lungs within 1–6 h and resulted in elevated levels of phospho-Smad1/5/8 [135] and reduction in vascular remodeling in the MCT-treated rats. The author speculated that the BMPR2-expressing BM-ELPCs exosomes enriched with BMPR2 could have had a protective effect on the pulmonary vasculature [136], suggesting exosome as an untapped pharmacological avenue for targeted delivery in PAH.

b.Ataluren/PTC124 Acting Via the Regulation of Ribosomal Translational Read-through

Approximately 75% of BMPR2 mutations linked to PAH are heterozygous nonsense mutations that are caused either by frame-shift deletions or insertions, whereby a premature translation codon (PTC) is inserted into the DNA sequence, resulting in an unstable mRNA transcript that is quickly degraded by nonsense-mediated decay (NMD) with complete absence of the mutant protein [137]. Nonsense mutations are quite common and account for almost 12% of all identified disease-causing mutations in the Human Gene Mutation Database [138]. When the mutation occurs closer to the 3’ end of a transcript, it could result in the transcript evading NMD, with a functional impact on the PTC that can be difficult to be predict [139]. Indeed, a large-scale RNA-Seq study suggested that approximately 68% of the variants presumed to be causing NMD were evading RNA surveillance [140].

Proof-of-concept investigations using repurposed aminoglycosides to correct nonsense mutations by inducing read-through of PTCs has so far mostly focused on diseases other than PH, such as, for example cystic fibrosis (CF) and Duchene muscular dystrophy (DMD) [141,142]. These antibiotics promote attachment to the ribosomal RNA and thereupon promote integration of an extra amino acid that subsequently results in a full-length protein [143,144]. Gentamicin was tested in a PAH model, where it was able to stimulate translation. The clinical use of the drug has, however, been limited by its oto and nepho toxicities [145]. Ataluren (PTC124), a non-aminoglycoside oxadiazole that improved read-through without major systemic side effects, was discovered in a high-throughput screen of compounds that affected the nonsense mutations [146]. Clinical trials in CF and DMD patients showed good efficiency and a favorable safety profile [147,148,149]. In a similar study, Drake et al. [149] observed PTC124’s ability to ameliorate the expression of BMPR2, BMP-induced signaling, and thereupon reverse proliferation in the vascular cells isolated from PTC-mutant BMPR2 patients. Similarly, Long et al. also conducted a pre-clinical study of PTC124 in PAH-patient-derived blood outgrowth endothelial cells (BOECs) and knock-in mouse model systems, both of which had the nonsense BMPR2 mutation, R584X [150]. The results from this study showed that in the mutant BOECs, PTC124 treatment increased both BMPR2 protein expression and ID1 signaling [150]. Additionally, PTC124 considerably improved endothelial vascular permeability, apoptosis, and proliferation, which have been described in association with BMPR2 mutations. Lastly, the oral administration of PTC124 augmented lung BMPR2 protein expression while reducing hyper proliferation in PASMCs isolated from mice with the R584X mutation. Nonetheless, PTC124 did not recover BMPR2 expression or BMP signaling in all PTC mutants [150]. Future pre-clinical studies probing the possibility of PTC124 treatment in patients carrying the precise nonsense BMPR2 mutation will be necessary before the drug can be considered therapeutically beneficial.

c.Tacrolimus/FK506 Acting Via the Indirect Cytoplasmic FKBP12 Blockade

Spiekerkoetter et al. [78] conducted a large-scale drug screen with 3756 FDA-approved drugs and bioactive compounds using a transcriptional high-throughput luciferase reporter assay to identify new state-of-the-art therapeutic modalities that could stimulate *BMPR2* signaling. They found that a low dose of the immunosuppressive drug tacrolimus (FK506) was a catalyst for BMP/SMAD signaling and reversed dysfunctional BMPR2 signaling. Tacrolimus (FK506) had a dual mechanism of action, stimulating BMP signaling while also inhibiting the interaction between the BMPR type-1 receptor kinase domain and the endogenous intracellular BMP signaling inhibitor FKBP12. [78]. In experimental PH models of endothelial Bmpr2 depletion, FK-506 improved PH. Similar results were found in a study that used IPAH patient-derived PMECs, showing that endothelial dysfunction could be rescued even in the presence of BMPR2 mutation or ablation [78]. The FK506 treatment also upregulated ALK1 and endoglin gene and protein expression in ECs [151].

Based on this data, two clinical trials were piloted to test the efficiency of FK506 in PAH. In a small-sized case study, low-dose of FK506 ameliorated cardiac function and considerably improved BMPR2 gene expression in three patients with end-stage PAH [152]. Conversely, a larger patient cohort (23 subjects) Phase IIa trial revealed no dose response or difference in any of the clinical endpoints or the levels of total BMPR2 and ID1 between the groups [153]. Nonetheless, a subset of patients showed an association between improved RV function and levels of BMPR2 expression. The authors therefore concluded that treatment with low-dose FK506 (2 ng/mL) was safe but that a larger multi-center trial among BMPR2-deficient PAH patients is needed to establish efficacy [153].

d.Sotatercept/ActR2a-Fc directed towards the abnormal BMP/activin/growth and differential factors (GDF) signaling via cell membrane:

Another promising approach to PH targets in the imbalance in activin/GDF (growth and differential factors) and BMP signaling. Activin and GDF are both associated with the TGFb superfamily and activate SMAD2/3 signaling [154]. Circulating levels of activin A have been shown to be elevated in humans with PAH, and increased levels have been described in the lungs of mice with hypoxia-induced PH [155]. In line with this study, activin-A-defusing antibodies diminished the pro-proliferative phenotype of PAH PASMCs, supporting an activin-A imbalance [156]. Similarly, ActR2a expression was increased in MCT-treated rat PASMCs, while Bmpr2 expression was considerably decreased, indicating a significant association between these type 2 receptors [157]. Based on these findings, it was hypothesized that the increased expression of ActR2a might result from the dysregulated feedback caused by BMPR2 depletion. Supporting this hypothesis, when BMPR2 is deficient, TNFα significantly and aberrantly induces ActR2a expression [158]. Additionally, while ActR2a is the primary type 2 receptor in activin and GDF8/11 signaling, it can also transduce BMP9 signaling when combined with ALK1 [159].

A few recent reports suggested that implementing a ligand trap to target activin/GDF signaling could be a feasible therapeutic approach for PAH. To test this in both in vitro and in vivo PH model systems, Yung et al. utilized the ActR2a ECD-Fc fusion protein (ActR2a-Fc/Sotatercept). Sotatercept was able to effectively block the activin and GDF8/11 activation of SMAD2/3 and additionally weakened the proliferation of the PASMCs and PMECs, thereby recovering and re-stabilizing the activin/BMP relationship [160].

The clinical efficacy of Sotatercept has been evaluated in two open-label trials for treating anemia and multiple myeloma, where it appeared to be well-tolerated [161,162]. Additionally, a Phase II PAH trial (PULSAR; ClinicalTrials.gov (accessed on 8 November 2022), identifier NCT03496207) randomized 106 patients already on background PAH therapy to receive placebo or 0.3 or 0.7 mg/kg Sotatercept every 3 weeks for a 24-week period. Patients in the Sotatercept treatment arm had a decrease in PVR (>18% decrease) and improvement in the 6MWD (>24 m), with 97 of 106 patients now being enrolled in the SPECTRA long-term Phase IIa trial [163,164].

e.Etanercept Targeting Inflammation That affects *BMPR2* Via Cell Membrane

Inflammation is widely thought to be a cause of PAH. Indeed, inflammatory cells are commonly found in plexiform lesions [165], and numerous inflammatory mediators, such as macrophage inflammatory protein-1a, interleukin-1b, IL6, and the 5-lipoxygenase pathway, have been linked to PAH [166] and to lower survival [167]. Among these cytokines, the heightened IL6-receptor (IL6R) was associated with poor prognosis and pulmonary vascular remodeling in mice [168,169]. In addition, overexpression of IL6 receptor resulted in hyperproliferation of PASMC and inhibition of apoptosis, which was effectively reversed in animal models after treatment with ERBF (20 S, 21-epoxy-resibufogenin-3-formate), a non-peptide IL6R antagonist [170]. On the basis of these pre-clinical observations, a Phase II open-label trial with tocilizumab, a potent IL6R-counteracting antibody, was conducted (TRANSFORM-UK; ClinicalTrials.gov (accessed on 8 November 2022), Identifier: NCT02676947) [171]. Despite being sufficiently powered, the treatment regimen failed to meet the primary and secondary endpoints [172]. Nonetheless, there was a signal that the treatment had higher efficacy in patients with connective tissue disease (CTD)-associated PAH, where higher levels of IL6 have been described compared to other PAH etiologies, emphasizing the heterogeneity of the disease and the importance of differentiating patients within the PAH umbrella diagnosis [172,173].

This aside, the TNFα was also found to be increased in PAH [167]. Transgenic mice with TNFα overexpression in the lung developed PH and showed down-regulation of BMPR2 mRNA and protein expression in both ECs and PASMCs [158]. The TNFα inhibitors have been commonly used to treat diverse inflammatory diseases such as inflammatory bowel disease, rheumatoid arthritis (RA), psoriasis, psoriatic arthritis, and heart failure [174,175,176,177]. Etanercept, an FDA-approved TNFα receptor type-2 ECD-Fc fusion protein employed to treat RA and psoriasis, improved vascular remodeling and hemodynamics in MCT-treated rats while also decreasing PVR in pigs with acute endotoxin-induced PH [178,179]. Despite the fact that there are no clinical trials currently underway to investigate the effects of TNF inhibitors in PAH, and considering the relatively safe profile of this treatment approach, it is indeed a potential target for investigation in the near future.

f.Hydroxychloroquine Promoting Degradation and Autophagy Via Lysosome

Viral infections have been identified as a potential etiological agent in the development of PAH. Specifically, infection with human herpesvirus 8 (Kaposi sarcoma-associated herpes virus (KSHV)) was initially linked to PAH, but these findings were not replicated in other patient cohorts [180,181,182]. The KSHV expresses a membrane-bound RING E3 viral ubiquitin ligase, K5, that is known to infect ECs [183,184] and to down-regulate cell-surface *BMPR2* via K5 ubiquitination of the membrane-proximal lysine in the cytoplasmic domain, leading to lysosomal degradation; a process that can be halted by the lysosomal inhibitor, concanamycin A [185].

Chloroquine and hydroxychloroquine are lysosomal inhibitors that have been extensively utilized in the prevention of malaria and as anti-inflammatory therapeutic avenues for treating RA and systemic lupus erythematosus (SLE) [186]. The treatment regimen with chloroquine/hydroxychloroquine appears to have a dual mechanism of action by increasing BMPR2 signaling and blocking autophagy as a result of an increased lysosomal pH [187]. Chloroquine/hydroxychloroquine treatment prevented the development or progression of PH, restored *BMPR2* expression, and decreased abnormal vascular cell responses in MCT-treated rats, where there was an elevated level of autophagy [187]. Additionally, chloroquine inhibited the degradation of p62—a crucial protein that is diminished during autophagy—which increased cell-surface *BMPR2* expression in PAECs in vitro, thereby preventing the receptor’s rapid turnover and salvaging the expression after siRNA knockdown. It also improved the BMP9–BMPR2 signaling targets in BMPR2-mutant ECs and restored BMPR2 cell-surface expression in BOECs isolated from patients with BMPR2 mutations [188].

Gomez-Puerto et al. [189] utilized a *BMPR2* HaloTagVR system to confirm and validate the above findings, demonstrating how blocking the lysosome in PAECs and PASMCs improved the expression of total and plasma-membrane BMPR2 levels. Interestingly, the authors discovered that *BMPR2* heterozygosity and inflammation induced autophagy. And, since inflammation is one of the predisposing factors for PAH, a therapeutic regimen with hydroxychloroquine may offer additional benefits by affecting the inflammatory pathways [190].

To conclude, despite significant developments in the treatment of PAH over the last 40 years, patient life expectancy remains dismal. Combination therapy with prostacyclin analogues and other forms of vasodilator therapeutic regimen, coupled with the development of expert centers for the care of PAH patients, have ameliorated survival rates. Nonetheless, there is a substantial need for therapies that directly target the underlying molecular mechanisms. Among those, are treatment approaches that aim at the *BMPR2* signaling pathway that focus on a drug target which may stimulate novel therapeutic strategies in the coming years (Figure 8). Further studies with appropriate stratification of patient cohorts based on the etiology of PAH will be required to translate these findings to clinical/pre-clinical set-up (Table 2).

## Figures and Tables

**Figure 1 cells-11-03823-f001:**
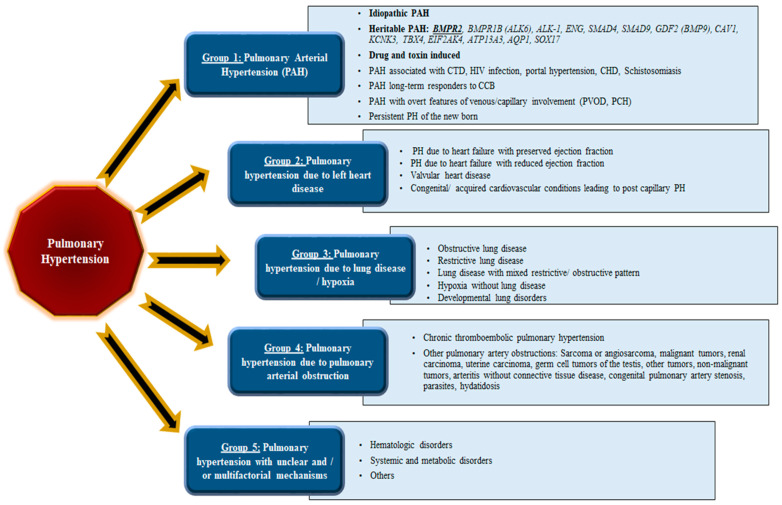
Updated clinical classification of pulmonary hypertension (PH), comprising five distinct groups based on recommendations from world experts at the most recent World Symposium on Pulmonary Hypertension.

**Figure 2 cells-11-03823-f002:**
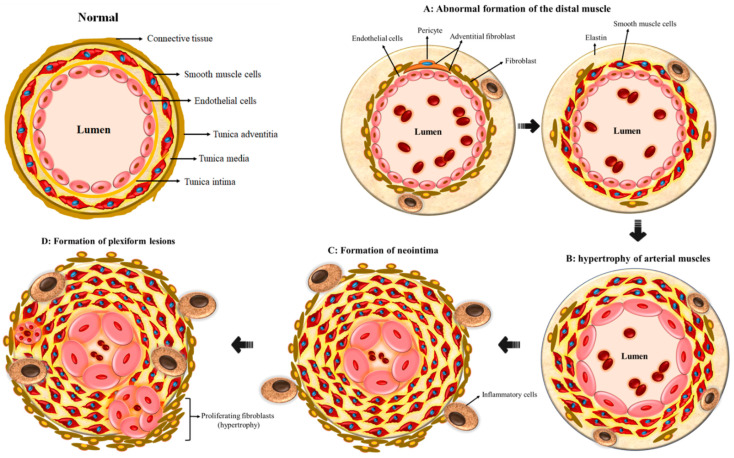
Schematic representation of pre-capillary arterieal remodeling in pulmonary arterial hypertension. (**A**) The condition commences with irregular distal muscle cell development trailed by (**B**) propagation of fibroblast and hypertrophy of the smooth muscle cells; (**C**) this primes the formation of neointima and plexiform lesions, subsequently leading to (**D**) constricted blood vessels that in turn increase the pulmonary blood pressure.

**Figure 3 cells-11-03823-f003:**
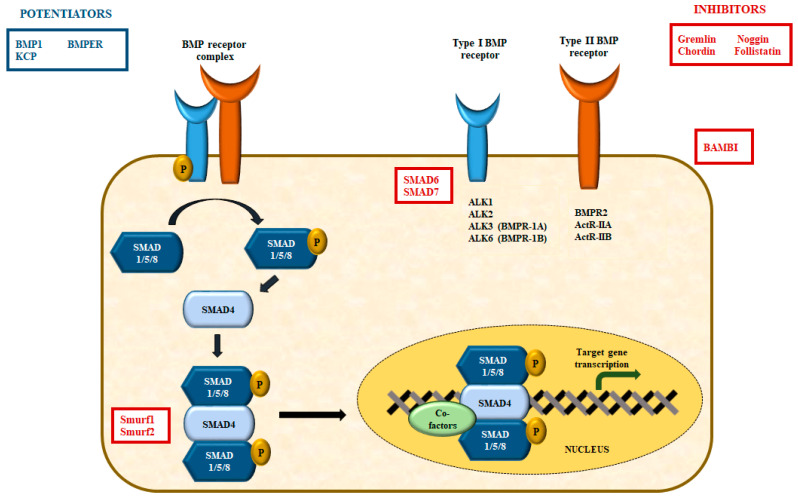
Summary of bone morphogenetic protein receptor signal transduction pathway. The catalysts or activators of BMP signaling are depicted in the blue box, while BMP signaling blocking agents or inhibitors are depicted in red boxes. BMP, bone morphogenetic protein; ALK, activin receptor-like kinase; BMPR; bone morphogenetic protein receptor; ActR, activin receptor; BMPER, BMP binding endothelial regulator; BAMBI, BMP and activin membrane bound inhibitor; KCP, kielin/chordin-like protein.

**Figure 4 cells-11-03823-f004:**
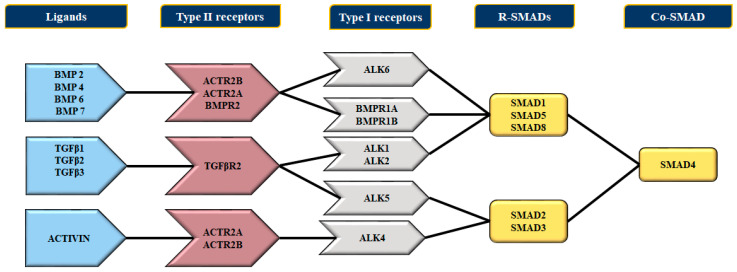
Diagrammatic representation portraying the correlation of BMP ligands, type II and type I receptors, and the SMAD proteins during signal transduction. Regarding the binding of BMPs to type I receptors, BMP-2 and BMP-4 bind to BMPR-IB receptors. The BMP-6 and BMP-7 bind robustly to ALK-2 receptors and less strongly to BMPR-IB receptors; BMP-9 and BMP-10 bind to ALK-1 and ALK-2, respectively. Similarly, the ALKs are classified depending on their structural and functional morphology and role. The ALK1 group comprises ALK1 and ALK2, whereas the BMPR1 group comprises BMPR1A/ALK3 and BMPR1B/ALK6. The receptors of the ALK1 group and BMPR1 group phosphorylate SMADSs 1/5/8, while the TGF-βRI group phosphorylates SMADs 2/3. The type II receptors are BMPR2 activin receptor 2A (ACTR2A) and 2B (ACTR2B). The BMPR2 exclusively binds BMPs, while ACTR2A and ACTR2B can bind to activin, myostatin, and BMPs. Type II receptors bind most BMP ligands and interfere with the binding of BMPs to type I receptors.

**Figure 5 cells-11-03823-f005:**
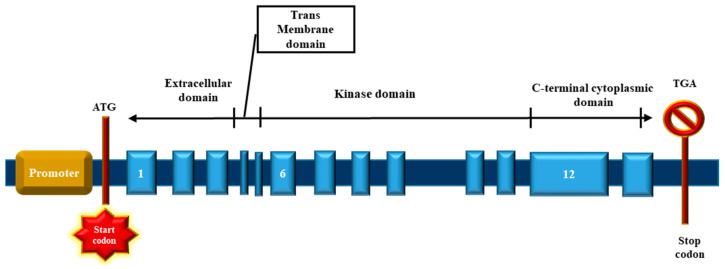
Diagrammatic representation of *BMPR2* gene. The *BMPR2* gene comprises of 13 exons that encodes the extracellular ligand-binding domain (exons 1 to 3), transmembrane membrane domain, (exons 4 to 5), kinase domain (exons 6 to 11), and the C-terminal tail domain (exons 12 to 13).

**Figure 6 cells-11-03823-f006:**
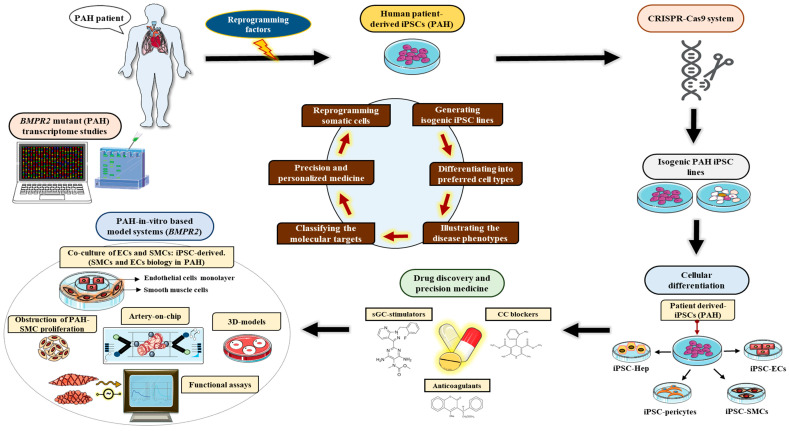
Human induced pluripotent stem cell (iPSC) utility as disease modeling systems. Patient-derived iPSCs, harboring disease-related mutations (i.e., PAH: BMPR2), can be “corrected” into control iPSCs by means of the CRISPR-Cas9 gene editing strategy to produce isogenic cells/lines. To replicate the PAH phenotype “in a dish”, the patient-derived iPSCs can be further differentiated into endothelial cells (ECs), smooth muscle cells (SMCs), hepatocytes (Hep), pericytes, and other related cell types. These specific differentiated cell types are then used with the 3D models described in the figure to study the underlying molecular and functional mechanisms that are responsible for specific cellular phenotypes. Such results can eventually pave the way for drug screening (soluble guanylyl cyclase: sGC; calcium channel blockers: CCB; and others) and precision medicine studies. Furthermore, the iPSC-derived hepatocytes hold great promise for testing drug screening and toxicity assays in a more customized and precise way.

**Figure 7 cells-11-03823-f007:**
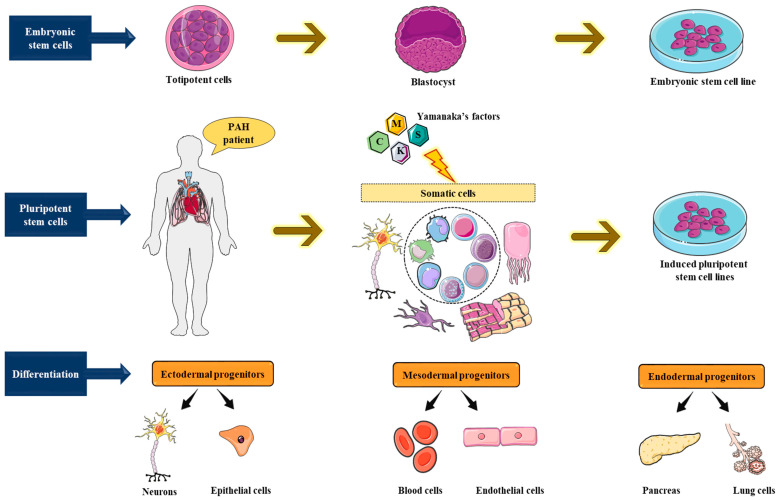
The unlimited potential of pluripotent stem cells in disease modeling. Embryonic stem cells (ESCs) and induced pluripotent stem cells (iPSCs) can perpetually self-renew and differentiate into multiple cell forms or schemes of the three embryonic germ layers viz., the ectoderm, mesoderm, and endoderm. The ESCs are derivatives of the inner cell mass of blastocyst stage embryos, while iPSCs are derived by means of nuclear reprogramming of somatic cells through the expression of a selected group of transcription factors, classically Oct4, Sox2, Klf4, and c-Myc.

**Figure 8 cells-11-03823-f008:**
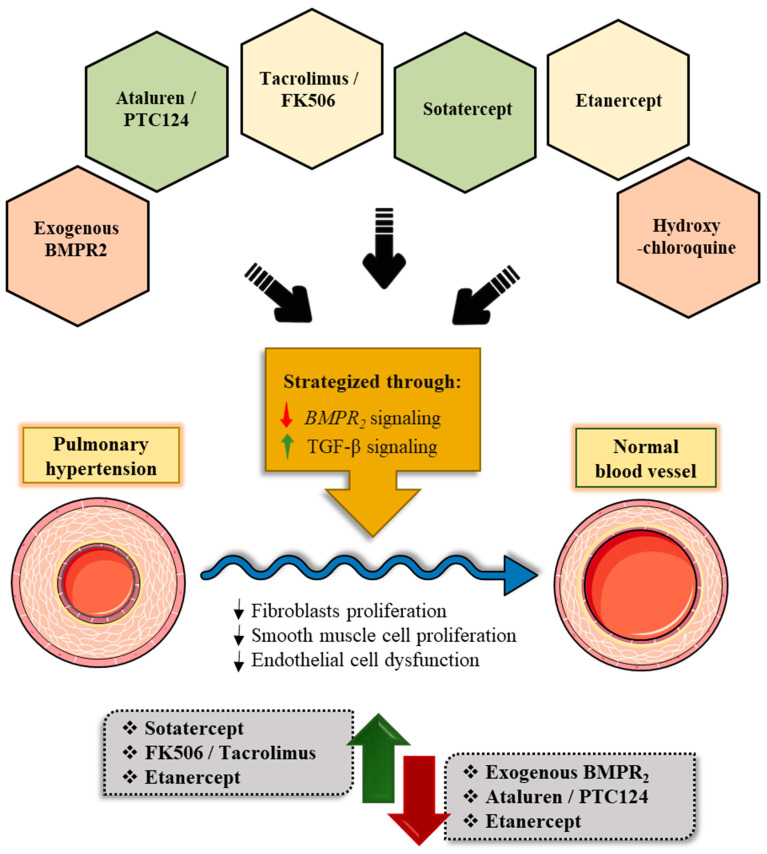
Emerging and existing therapeutic targets that ameliorate the expression of *BMPR2* and/or BMP signaling. A better understanding of the molecular and genetic components in PAH has opened new avenues for the development and testing of novel therapeutic modalities. Increasing stability of the BMPR2/BMP and TGFb pathways will be essential for maintaining normal vascular homoeostasis. Manipulation of these pathways will be crucial for reducing/reversing the vascular remodeling that is linked to PAH pathogenesis.

**Table 2 cells-11-03823-t002:** Overview of prospective therapeutic drug target steering via BMPR2 and BMP signaling in pulmonary arterial hypertension (PAH).

Drug/Compound	Route of Administration	Mechanism of Action	Clinical Trial	Pre-Clinical Animal Studies	Pre-Clinical Human Studies	References
Ataluren/PTC124 (Translama)	Oral	Regulates translational read-through of premature stop codons in BMPR2 mutants	--	Bmpr2 R584X mice	PAECs and BOECs, Bmpr2 R584X PASMCs	[149,150,191]
Exogenous BMPR2		Delivery of wild-type BMPR2 via viral like particles (VLP), exosomes, or nanoparticles	--	Chronic hypoxia rats; Chronic hypoxia and MCT rats	Non-transformed mouse mammary gland epithelial cells (NMuMGs) and human umbilical vein endothelial cells (HUVECs), PMECs, BM-ELPCs	[132,133,134,135]
BMP9/10		Regulates BMP signaling and improved expression of BMPR2		MCT rats, SU-5416 hypoxia rats, and Bmpr2 R899X mice	PAECs and BOECs	[159,192]
Etanercept (Embrel)	Subcutaneous injection	Obstructs TNF-alpha-induced inflammation and the downregulation of BMPR2 by performing as a TNF-alpha ligand trap		MCT rats; endotoxaemic pigs; SU-416 hypoxia rats	PAECs and PASMCs	[158,175,179,193]
Tacrolimus/FK506	Oral	Reestablishes BMP signaling by the inhibition of FKBP12 signaling blockage.	Low-dose FK506 at end-stage PAH; transform PAH— NCT01647945 at Phase II	MCT rats, SU-5416 hypoxia rats, and EC-Bmpr2^−/−^ mice	PAECs	[78,152,153]
Sotatercept (ACE-011)	Subcutaneous injection	Inhibits TGF beta signaling by serving as an activin ligand trap	PULSAR—NCT03496207 (Phase II) SPECTRA—NCT03738150 (Phase IIa)	MCT rats and SU-5416 hypoxia rats	PMECs and PAMSCs129	[160,161,162,194]
Hydroxychloroquine sulfate (Plaquenil)	Oral	Inhibits the lysosomal damage of BMPR2		MCT rats	PASMCs and PAECs; PAECs and BOECs; PAECs, PASMCs, and HMEC	[186,187,188,189]
Olaparib (Lynparza)	Oral	Prevents PARP1-induced DNA restoration in the deficiency of BRCA1 expression	Olaparib for PAH— NCT03782818 (Phase Ib)	MCT rats and SU-5416 hypoxia rats	PASMCs	[195,196]
Sodium 4-phenylbutyrate/4PBA or glycerol phenyl butyrate (Ravicti)	Oral	Discharges endoplasmic-reticulum-confined BMPR2		Bmpr2 ΔEx2 mice; Bmpr2 C118W mice; chronic hypoxia mice; chronic hypoxia mice and MCT rats	HeLas and MRC-5s; Bmpr2 ΔEx2 PMECs; BMPR2 C118W HDFs and Bmpr2 C118W PASMCs	[191,197,198,199,200,201]

PAECs: pulmonary artery endothelial cells; BOECs: blood outgrowth endothelial cells; PASMCs: pulmonary artery smooth muscle cells; NMuMGs: non-transformed mouse mammary gland epithelial cells; PMECs: pulmonary microvascular endothelial cells; HUVECs: human umbilical vein endothelial cells; BM-ELPCs: bone-marrow-derived endothelial-like progenitor cells; HMEC: human dermal microvascular endothelial cells; HDFs: human dermal fibroblasts; HeLa: Henrietta Lacks (uterine cell variety; named for deceased patient); MCT: monocrotaline; VLP: viral-like particles.

## Data Availability

No new data were created or analyzed in this study. Data sharing is not applicable to this article.

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
