# Peer review of "The Role of Bone Morphogenetic Protein Receptor Type 2 (BMPR2) and the Prospects of Utilizing Induced Pluripotent Stem Cells (iPSCs) in Pulmonary Arterial Hypertension Disease Modeling"

_cells, 2022, doi:10.3390/cells11233823_

Round 1

Reviewer 1 Report (Previous Reviewer 1)

The authors have adequately revised the article, which can be improved upon after a thorough review in English

Author Response

We thank the reviewers and the editorial board for reviewing our manuscript and for their helpful and insightful comments. Below, and in the revised manuscript, we have addressed each of these points. In addition, we have highlighted the modification in the main manuscript in red.

  • The authors have adequately revised the article, which can be improved upon after a thorough review in English.

We have made an extensive review of the English language utilized in the paper and made changes accordingly.

Reviewer 2 Report (Previous Reviewer 3)

Thank you for the response to my comments. Regarding the BMPR2 mutations, it seems unclear to me as to why the information on the BMPR2 mutations has not actually been included in the manuscript. Would this not be meaningful information for the reader to know? If the authors believe that available studies are too small in number to draw conclusions regarding genotype phenotype correlations then this could be added as a limitation. Please note that additional iPSC lines have been generated (eg PMID: 34388490) but they have not been referenced here.       

Author Response

We thank the reviewers and the editorial board for reviewing our manuscript and for their helpful and insightful comments. Below, and in the revised manuscript, we have addressed each of these points. In addition, we have highlighted the modification in the main manuscript in red.

  • Thank you for your response to my comments. Regarding the BMPR2 mutations, it seems unclear to me as to why the information on the BMPR2 mutations has not actually been included in the manuscript. Would this not be meaningful information for the reader to know? If the authors believe that available studies are too small in number to draw conclusions regarding genotype-phenotype correlations then this could be added as a limitation.

We have addressed this comment pertaining to the information with regard to the phenotype-genotype correlation and have added it as a limitation of this review paper (please refer to page 18, lines 364 – 367)

  • Please note that additional iPSC lines have been generated (eg PMID: 34388490) but they have not been referenced here

We have addressed this comment and made the necessary changes and have included this paper (PMID: 34388490): please refer to page 17, lines 317 – 319.

Round 2

Reviewer 2 Report (Previous Reviewer 3)

Thank you for your response, I have no further comments.

This manuscript is a resubmission of an earlier submission. The following is a list of the peer review reports and author responses from that submission.

Round 1

Reviewer 1 Report

the review is very well written, and is well organized. I suggest adding a section dedicated to therapeutic strategies (pharmacological and non-pharmacological) acting through the BMPR2 pathway (including the recent sotatercept).

Author Response

We thank the Editors and the Reviewers for their thorough and thoughtful evaluation of our manuscript and their comments. The following sections contain a detailed response to the Editors’ and Reviewers’ remarks.

Authors' Responses to Reviewer's Comments (Reviewer 1)

“Comments and Suggestions for Authors: the review is very well written, and well organized. I suggest adding a section dedicated to therapeutic strategies (pharmacological and non-pharmacological) acting through the BMPR2 pathway (including the recent sotatercept)”

We thank the reviewer for this insightful comment. We have added a separate section under a new sub-heading (section 9) on the “Emerging therapeutic modalities for PAH targeting the BMPR2 signaling pathway”, where we briefly discuss novel therapeutic approaches targeting the BMPR2 signaling pathway.

Reviewer 2 Report

This manuscript reviewed the studies of PAH and PH, most focusing BMPR2 mutation and the in vitro model using patient iPSC. The manuscript described these in detail, which may be understandable for most of the readers who may have interest in this area. However, the figures of iPSCs in this manuscript are general not specific to PAH study. It may be better if the authors create the illustration specific to PAH study, for example, the in vitro 3D model or chip model of pulmonary endothelial cells and smooth muscle cells. Following are the several comments.

Page 10, Line 134, the authors stated that the animal models do not accurately reflect human diseases. I hope the authors describe the differences more specifically.

Page 11, Line 196, delete newly (duplicated).

Page 11, Lines 196-201, although I can understand what the authors would like to describe, the sentences are too complicated. Please describe in more clearer way. Also, period is missing.

Page 13, Figure 7, the authors explain the abbreviations like HPSC and MSC, which are not appeared in the figure.

Page 15, Line 89, why there is an apostrophe after smooth muscle cells?

References have 132 references, however, only 121 references were referred in the text.

Author Response

We thank the Editors and the Reviewers for their thorough and thoughtful evaluation of our manuscript and their comments. The following sections contain a detailed response to the Editors’ and Reviewers’ remarks.

Authors' Responses to Reviewer's Comments (Reviewer 2)

This manuscript reviewed the studies of PAH and PH, most focusing on BMPR2 mutation and the in vitro model using patient iPSC. The manuscript described these in detail, which may be understandable for most of the readers who may have an interest in this area. However, the figures of iPSCs in this manuscript are general, and not specific to the PAH study. It may be better if the authors create an illustration specific to the PAH study, for example, the in vitro 3D model or chip model of pulmonary endothelial cells and smooth muscle cells.

We thank the reviewer for this insightful comment. We have now revised the figures to make them more specific to PAH and describe both the 3D and chip model systems.

Page 10, Line 134, the authors stated that the animal models do not accurately reflect human diseases. I hope the authors describe the differences more specifically.

We have described the differences and made the required changes.

Page 11, Line 196, delete newly (duplicated).

This has been addressed.

Page 11, Lines 196-201, although I can understand what the authors would like to describe, the sentences are too complicated. Please describe it in a clearer way. Also, the period is missing

The sentence has been revised.

Page 13, Figure 7, the authors explain the abbreviations like HPSC and MSC, which do not appear in the figure.

This has been addressed.

Page 15, Line 89, why there is an apostrophe after smooth muscle cells?

We have made the corrections accordingly.

References have 132 references, however, only 121 references Ire referred to in the text.

The remaining references are in the tables.

Reviewer 3 Report

The manuscript by Devendran and co-authors describes the pathophysiology and genetic factors of pulmonary arterial hypertension. Emphasis is given to BMPR2, the most frequently mutated gene in PAH patients. The authors provide an overview of iPSC-based PAH disease modeling, an approach that has been employed in order to overcome the limited access to human tissue material and failure of animal models to accurately mimic the pathogenesis of human disease.  Overall, the manuscript is well written and has some nice illustrations.

Comments:

1) Although the title of the review is ‘iPSCs-based disease modeling’, the majority of the text is focused on BMPR2 rather than iPSc-modeling of PAH. The latter is only described in section 7 which is just over 1-page long in this 16-page manuscript. The authors should either change the title or revise the manuscript by condensing sections 1-6 and expanding section 7.

2) Given that so far iPSC studies have failed to recapitulate key patient phenotypic characteristics, are the authors optimistic about their use as a disease model for PAH? This is especially when keeping in mind the complexity of the disease and participation of cells of different origin.

3) In Figure 2, please enlarge the font as at its current form it is quite difficult to read. Also, not all cell types are explained in the figure (eg green with yellow nucleus). As an alternative way to annotate the different cell types, maybe a key can be included on the side of the figure showing each cell type and its annotation.

4) In Table 1, it may be helpful if the number or % of mutations per gene are included.

5) In Figure 5, the significance of panels C- E is unclear as they are not described anywhere in the main text. If these panels are not essential, please remove. Panel A seems redundant too.   

6) In the iPSC studies described in section 7, were all these studies performed on cells derived from patients with BMPR2 mutation? This information is mentioned for some studies but not all. If so, did they have the same BMPR2 mutation? Is there any genotype-phenotype correlations?   

7) Since the disease is clinically manifested in only 20% of BMPR2 carriers, how do the authors envision that iPSCs could help understand the mechanisms contributing to whether a carrier will be symptomatic or asymptomatic?

Author Response

We thank the Editors and the Reviewers for their thorough and thoughtful evaluation of our manuscript and their comments. The following sections contain a detailed response to the Editors’ and Reviewers’ remarks.

“Although the title of the review is ‘iPSCs-based disease modeling’, the majority of the text is focused on BMPR2 rather than iPSC-modeling of PAH. The latter is only described in section 7 which is just over 1-page long in this 16-page manuscript. The authors should either change the title or revise the manuscript by condensing sections 1-6 and expanding section 7

This has been addressed and the title revised.

Given that so far iPSC studies have failed to recapitulate key patient phenotypic characteristics, are the authors optimistic about their use as a disease model for PAH? This is especially when keeping in mind the complexity of the disease and the participation of cells of different origins.

We agree with the reviewer. The use of iPSCs and the development of 3D human multi-tissue constructs for disease modeling is still in its infancy, yet we believe quite promising. As described in the review, endothelial cells (ECs), smooth muscle cells (SMCs), and pericytes can be generated from iPSCs using cutting-edge protocols to produce cells that are more and more similar to pulmonary vascular cells. Additionally, the co-culture of iPSC-derived ECs and perivascular cells might recapitulate more closely the highly specialized and complex nature of the pulmonary vascular. We recognize that animal models have been widely used to model diseases, yet they are costly, time-consuming, and frequently fail to fully replicate human pathology. iPSC derives multi-tissue structure and even Organoids, which are derived from iPSCs, could eventually replace animal models.

In Figure 2, please enlarge the font as in its current form it is quite difficult to read. Also, not all cell types are explained in the figure (eg green with a yellow nucleus). As an alternative way to annotate the different cell types, maybe a key can be included on the side of the figure showing each cell type and its annotation.

This has been addressed.

In Table 1, it may be helpful if the number or % of mutations per gene is included.

We prefer to leave it as is given that specific details are available primarily for the BMPR2 mutation.

 In Figure 5, the significance of panels C- E is unclear as they are not described anywhere in the main text. If these panels are not essential, please remove them. Panel A seems redundant too.  

We appreciate this comment but we aimed at being comprehensive. Unless the reviewer feels strongly about it, we would prefer to leave Figure 5 as is.

In the iPSC studies described in section 7, are all these studies performed on cells derived from patients with BMPR2 mutation? This information is mentioned in some studies but not all. If so, did they have the same BMPR2 mutation? Are there any genotype-phenotype correlations? 

Unfortunately, there were also differences in the genotype and the number of available studies is quite small to make meaningful comparisons. Standardization of the protocols and the use of de-novo mutant lines with a matching background might address these relevant concerns.

 Since the disease is clinically manifested in only 20% of BMPR2 carriers, how do the authors envision that iPSCs could help understand the mechanisms contributing to whether a carrier will be symptomatic or asymptomatic?

In the manuscript, we referenced the paper (ref. 71) published by the Rabinovitch group in Standford, where they describe molecularly and functional differences in the properties of iPSC-EC derived from carriers of the BMPR2 mutation with and without PAH. We infer that this might be a potential strategy to address this important question.

Round 2

Reviewer 2 Report

I do not have more comments.

Reviewer 3 Report

Thank you for the detailed point-to-point response to my comments. However, it appears that the majority of my comments have not been addressed appropriately and minimal revisions have been applied to the text. For example, despite author’s reply on comment-1 that the title have been changed to reflect the fact that the manuscript is not focused on ‘iPSCs-based disease modeling’, this is not the case. If the authors believe that all panels of Figure 5 are necessary, then they should at least be mentioned and explained in the main text of the manuscript. Also, regarding my questions about the iPSC studies mentioned in section 7 and whether these are all on BMPR2 mutations and if so whether these are the same mutations, no reply has been provided. This information should be available in the corresponding references so should be feasible to address this comment.

Author Response

We thank the Editors and the Reviewers for their thorough and thoughtful evaluation of our manuscript with their invaluable comments, inputs, and subject expertise. The following sections contain a detailed response to the Editors’ and Reviewers’ remarks.

We sincerely apologize for not addressing a few of the comments addressed by reviewer 3.

We have made the appropriate changes and revisions as pointed out as follows:

  1. We have modified the title of the review article to “The role of bone morphogenetic protein receptor type 2 (BMPR2) and the prospects of utilizing induced pluripotent stem cells (iPSCs) in Pulmonary arterial hypertension disease modeling”.
  2.  We have e revised figure 5, as suggested by the reviewer.
  3.  The studies that we had discussed in “section 7”, which we have now modified as “section 6.1”, are chiefly from both patients (IPAH, HPAH: BMPR2 c.354 > G; BMPR2 c. 250delC; BMPR2 c.1471C >T; p.R491W) and healthy controls (BMPR2 c.G350A) and also from mouse embryoid bodies. However, most of these references do not provide information about the BMPR2 mutation status. And therefore, with the prevailing number of available studies, there are also differences in the genotype. The number of available studies is relatively small to make meaningful comparisons at this point.
